# Task-Adaptive Pretrained Language Models via Clustered Importance Sampling

**David Grangier, Simin Fan, Skyler Seto, Pierre Ablin**
Apple

## Abstract

Specialist language models (LMs) focus on a specific task or domain on which they often outperform generalist LMs of the same size. However, the specialist data needed to pretrain these models is only available in limited amount for most tasks. In this work, we build specialist models from large generalist training sets instead. We propose a novel method, ClusteRed Importance SamPling (CRISP). CRISP clusters the generalist dataset and samples from these clusters based on their frequencies in the smaller specialist dataset. It is scalable, suitable for both pretraining and continued pretraining, and works well in multi-task settings. CRISP performs favorably compared to other methods that adjust the training distribution of the generalist data with guidance from the limited domain-specific data. Our findings demonstrate improvements across different domains in terms of language modeling perplexity and accuracy on multiple-choice question tasks. We also present ablation studies that examine the impact of dataset sizes, clustering configurations, and model sizes.

## 1 Introduction

Generalist language models (LMs) can address a wide variety of tasks, but this generality comes at a cost (Brown et al., 2020). It necessitates a large training set representative of all prospective tasks, as well as a large model to fit such a comprehensive dataset. Specialist models forgo this generality and fit a model for a limited domain or task. In their narrow specialty, such models can achieve better accuracy at a given model size (Kerner, 2024).

Pretraining a specialist is interesting when two conditions are met: (i) the targeted task justifies the cost of training a dedicated model and (ii) a specialist dataset large enough for pretraining is available. Condition (i) is dependent on the targeted application and its potential economic benefit. Condition (ii) is more limiting since modern LMs are commonly pre-trained on datasets larger than 100B tokens[1], an amount that cannot be commissioned for most applications.

This work considers relaxing condition (ii) and studies methods to train a specialist model when specialized data is scarce. Given a large generalist dataset and a small specialist dataset, we propose to modify the distribution over the generalist dataset guided by the scarce specialist dataset. Training a model on the modified distribution gives a specialist model with better accuracy than a generalist model of the same size.

We study this setting across different specialization tasks including domain-specific language modeling (medical, encyclopedic domains) and end-tasks (scholar exams in science and humanities, reasoning questions). We compare different strategies to manipulate the pretraining distribution. We evaluate strategies based on text classifiers, gradient-alignment and importance sampling (IS). Although IS is rarely used for LM data selection, we build upon on a simple IS recipe based on clustering (Grangier et al., 2024b) and report that the resulting method systematically outperforms alternatives. Our IS recipe clusters the generalist set and computes the cluster histogram over the specialist data. Then, for pretraining, generic data is sampled according to the specialist histogram, see Figure 1. We show the empirical benefit of this method varying model sizes (350m to 7B parameters), the amount of generalist data and the amount of specific data. We assess both perplexity gains for language model adaptation and accuracy improvements for multiple choice question tasks.

---

[1] 100B tokens $\simeq$ 1m books $\simeq$ 60x the annual publication of the top English language publisher (Lee, 2021).

This paper presents an exhaustive comparison over different model sizes (350m, 1.3B, 6.8B) and different numbers of clusters (scaling from 64 to 16m clusters with hierarchical clustering). We consider different tasks, both for language modeling and multiple-choice questions. We also explain the impact of hyperparameters such as the clustering representation and number of clusters. We study IS in the context of multitasking and continued pretraining. We also perform ablations with respect to the generic pre-training set size and the specialization data size.

## 2 RELATED WORK

**Generalist vs Specialist LMs** Generalist LMs address tasks for which they have not been explicitly trained (Brown et al., 2020) or provide a good initialization for fine-tuning a dedicated model (Devlin et al., 2019). Nowadays generalists compete with dedicated models on many tasks (Jiang et al., 2024; Dubey et al., 2024). Success, however, comes at a price: a generalist must be much larger than a specialist for the same accuracy. For instance, on English-to-German translation, the 175-B parameter generalist GPT-3 (Brown et al., 2020) is less accurate than a 136m-parameter specialist (Sennrich et al., 2016a). For neural LMs, the parameter count directly impacts training and inference costs.

Specialist large LMs exist in domains where large amounts of specialized texts are available. Corpora with billions of tokens enable pretraining or *continued pretraining*, a generalist pretraining phase followed by a specialist one (Gururangan et al., 2020; Parmar et al., 2024)). Domains with specialist models include medicine and biology (Lewis et al., 2020; Labrak et al., 2024; Bolton et al., 2024), computer programming and mathematics (Lewkowycz et al., 2022; Rozière et al., 2024; Azerbayev et al., 2024) and finance (Wu et al., 2023; Xie et al., 2023a). When specialist data is available in limited amount, task-adaptive data-selection methods train specialist models on generalist data instead.

**Task-Adaptive Data-Selection** These selection methods over-sample generalist data that aids model generalization in the target domain. For masked LMs, Gururangan et al. (2020) observe that continued pretraining improves the performance on end-tasks when using data with high vocabulary overlap with the targeted task. For machine translation (MT), Aharoni & Goldberg (2020) show that a task-adapted pretraining dataset can be selected from a generalist dataset using the nearest neighbors of a small specialist set. Their nearest neighbor classifier relies on BERT sentence distance (Devlin et al., 2019). Still for MT, other works have used other types of classifiers. In particular, contrasting the scores of two LMs (generalist and specialist) is popular (Moore & Lewis, 2010; Axelrod et al., 2011; Wang et al., 2018; Junczys-Dowmunt, 2018). Other classifiers include logistic regression or fine-tuned BERT (Iter & Grangier, 2021). Outside classification, Xie et al. (2023c) proposed to use importance sampling for continued pretraining. They improve classification tasks by selecting pretraining data with a similar distribution to the targeted domain in terms of hashed-ngrams. Importance sampling is also used in (Grangier et al., 2024b) and we build upon that work which adjusts the frequency of generalist clusters informed by specialist data: we scale the method to millions of clusters, show that it works with larger models, and extend it beyond language modeling tasks.

A third type of methods for task-adaptative selection relies on bilevel optimization and gradient aligment (Pruthi et al., 2020; Xia et al., 2024; Grangier et al., 2023). The pretraining distribution is selected such that the reweighted gradients from the generalist dataset mimics the expected gradient from the small specialist dataset. Gradient-alignment for data selection has also been used for other purposes such as data summarization (Borsos et al., 2024), pretraining acceleration (Xie et al., 2023b; Fan et al., 2024) or auxiliary task weighting (Wang et al., 2020; Raghu et al., 2021). Finally, it is also worth mentioning data selection methods based on reinforcement learning (Liu et al., 2019; Yoon et al., 2020), bayesian optimization (Ruder & Plank, 2017), data models (Ilyas et al., 2022) and influence models (Yu et al., 2024).

**Pretraining Data Quality** Outside of domain aspects, the quality of pretraining data is also an important topic (Wenzek et al., 2020; Dodge et al., 2021; Penedo et al., 2023; Li et al., 2024). Data quality includes removing data in other languages (Cook & Lui, 2012), text formatting (Xu et al., 2024), favoring long form text Gao et al. (2021); Gunasekar et al. (2023), removing duplicates (Lee et al., 2022). It also involves balancing different sources of data with the goal of reaching a better generic pretraining loss Xie et al. (2023b); Fan et al. (2024); Vo et al. (2024). Recent work also considered filtering Kong et al. (2024), correcting Chen & Mueller (2024) or generating Maini

et al. (2024) pretraining data with LMs. These data quality considerations are orthogonal to domain concerns: quality filters are applied alongside domain adaptation decisions (Albalak et al., 2024).

## 3  DATA SELECTION FOR TASK-ADAPTIVE PRETRAINING

We consider three methods for task-adaptive pretraining of LMs. Classification and gradient alignment have been evaluated in different contexts before but not for end-tasks like multiple-choice question answering. Clustered-based importance sampling at scale is a contribution of this work, building upon recent work from Grangier et al. (2024b).

### 3.1  NOTATIONS

$D^{\mathrm{g}}$ is the training dataset sampled from the generalist distribution $\mathcal{D}^{\mathrm{g}}$. $D^{\mathrm{s}}$ is the specialist dataset representative of the final task, sampled from the specialist distribution $\mathcal{D}^{\mathrm{s}} \neq \mathcal{D}^{\mathrm{g}}$. The loss of model $\theta$ on a dataset $D$ is

$$\mathcal{L}(D;\theta) := \frac{1}{|D|} \sum_{x \in D} \ell(x;\theta) = -\frac{1}{|D|} \sum_{x \in D} \frac{1}{|x|} \sum_{i} \log p(x_i|x_1^{i-1};\theta)$$

where $|D|$ denotes the cardinality of $D$ and $|x|$ denotes the length of sequence $x = (x_1, \ldots, x_{|x|})$. The perplexity of model $\theta$ on the dataset $D$ is $\mathcal{P}(D;\theta) := \exp(\mathcal{L}(D;\theta))$.

### 3.2  CLASSIFICATION

A binary classifier is trained to estimate the probability that a generalist pretraining document belongs to the targeted domain. The classifier $\phi$ is learned using positive examples from $D^{\mathrm{s}}$ and a subset of $D^{\mathrm{g}}$ as negative examples. $\phi$ then builds a domain-specific pretraining set

$$C(D^{\mathrm{g}}, t) := \{x \in D^{\mathrm{g}} \text{ such that } \phi(x) > t\}.$$

which restricts the generic dataset $D^{\mathrm{g}}$ to the examples with an estimated probability to be in-domain above threshold $t$. The threshold $t$ is a sensitive hyperparameter that impacts the downstream model. It is validated as a trade-off between focusing on data close to the domain of interest while keeping $C(D^{\mathrm{g}}, t)$ large enough to train an LM of the targeted capacity. In our case, we rely on a logistic regression classifier trained over sentence BERT (SBERT) text embeddings (Reimers & Gurevych, 2019), an established classification method (Minaee et al., 2021). The SBERT representation is also commonly used in data selection (Albalak et al., 2024; Xie et al., 2023c; Zhang et al., 2024; Su et al., 2023). This representation is also used in the alternative selection strategies we consider. As an ablation, we also evaluate the impact of the choice of SBERT (Section 5.1).

### 3.3  GRADIENT-ALIGNMENT

Gradient-Alignment (GA) methods are common when the generic pretraining set $D^{\mathrm{g}}$ originates from $n_{\mathrm{g}}$ different data sources $S$, i.e. $D^{\mathrm{g}} = \bigcup_{i=1}^{n_{\mathrm{g}}} D_i^{\mathrm{g}}$. These methods select weights for the different sources by considering two functions of $\theta$: the pretraining reweighed loss,

$$\mathcal{L}((w, D^{\mathrm{g}}); \theta) := \sum_{i=1}^{n_{\mathrm{g}}} w_i \mathcal{L}(D_i^{\mathrm{g}}; \theta),$$

and the targeted loss, i.e. the loss on $D^{\mathrm{s}}$ in our case. The weights, on the simplex, can be inferred via a bilevel formulation of the data selection problem (Dagréou et al., 2022): the minimum $\theta^{\star}(w) = \arg\min_{\theta} \mathcal{L}((w, D^{\mathrm{g}}); \theta)$ depends on $w$ and task-dependent pretraining is interested in weights $w$ that minimize $\mathcal{L}(D^{\mathrm{s}}; \theta^{\star}(w))$ wrt $w$. This formulation results in algorithms that select weights during pretraining to align the gradients of these two functions wrt $\theta$ (Xie et al., 2023b; Grangier et al., 2023). In our case, we rely on the DoGE (Fan et al., 2024) algorithm. Compared to classifiers, GA is harder to scale to large model size. This limitation is commonly addressed by finding the mixture weights with a small model before transferring them to a larger model.

In this work, we consider a generic setting where the pretraining dataset $D^{\mathrm{g}}$ is not pre-segmented into few data sources. Instead, we rely on the k-means clustering of the Sentence BERT embeddings to identify data clusters. Clustering based on text embeddings has been used for data selection, both for quality filtering (Kaddour, 2023) and domain adaptation (Grangier et al., 2024a).

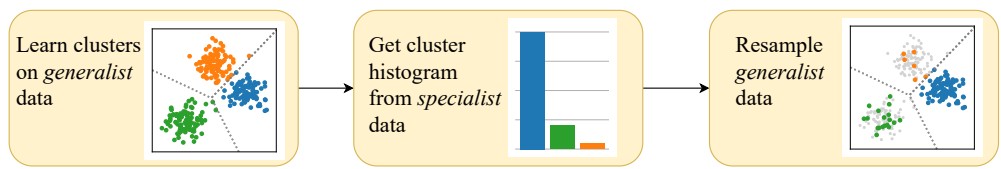

Figure 1: **Task-adaptive data selection with Clustered Importance Sampling (CRISP).**

### 3.4 CRISP: ClusteRed Importance Sampling for Pretraining

We sketch our strategy from Figure 1. Initially, we divide the space of text into clusters. We decompose the specialist loss and the generalist loss as a weighted sum of losses over clusters. Then we make an independence assumption that implies that the specialist and generalist loss per cluster are identical. The specialist loss is then computed as the generalist loss with a reweighing of each cluster.

Specifically, we want to identify a model with a low loss on the specialist distribution $\mathcal{D}^s$,

$$\mathcal{L}(\mathcal{D}^s; \theta) = \mathbb{E}_{x \sim \mathcal{D}^s}[\ell(x; \theta)] = \sum_x \ell(x; \theta) P(x|\mathcal{D}^s)$$

We marginalize over a discrete latent variable $c$, the cluster variable, and write

$$\mathcal{L}(\mathcal{D}^s; \theta) = \sum_x \sum_c \ell(x; \theta) P(x|c, \mathcal{D}^s) P(c|\mathcal{D}^s) \underset{(2)}{=} \sum_x \sum_c \ell(x; \theta) P(x|c) P(c|\mathcal{D}^s) \quad (1)$$

where the second equality $=_{(2)}$ makes the independence assumption $P(x|c, \mathcal{D}^s) = P(x|c)$. If we make a similar assumption for the generalist loss $P(x|c, \mathcal{D}^g) = P(x|c)$, we can write both losses as

$$\mathcal{L}(\mathcal{D}^s; \theta) = \mathbb{E}_{c \sim (c|\mathcal{D}^s)}[\mathcal{L}(c; \theta)] \quad \text{and} \quad \mathcal{L}(\mathcal{D}^g; \theta) = \mathbb{E}_{c \sim (c|\mathcal{D}^g)}[\mathcal{L}(c; \theta)] \quad (2)$$

where we define $\mathcal{L}(c; \theta) =: \sum_x \ell(x; \theta) P(x|c)$. We now apply importance sampling to these expectation, defining the importance weights as $w(c) = P(c|\mathcal{D}^s)/P(c|\mathcal{D}^g)$,

$$\mathcal{L}(\mathcal{D}^s; \theta) = \sum_c \mathcal{L}(c; \theta) P(c|\mathcal{D}^s) = \sum_c \mathcal{L}(c; \theta) \frac{P(c|\mathcal{D}^s)}{P(c|\mathcal{D}^g)} P(c|\mathcal{D}^g) = \mathbb{E}_{c \sim (c|\mathcal{D}^g)}[w(c)\mathcal{L}(c; \theta)].$$

In our experiments, we estimate the terms $w(c), \mathcal{L}(c; \theta)$ from the finite training sets $D^s \sim \mathcal{D}^s$ and $D^g \sim \mathcal{D}^g$. We count the number of examples in each cluster to estimate $P(c|D^s), P(c|D^g)$. The expected loss over a cluster $\mathcal{L}(c; \theta)$ is estimated as the average loss over the generalist examples in cluster $c$, $\mathcal{L}(D^g \cap K(c); \theta)$, where $K(c)$ denotes the examples in cluster $c$. This strategy therefore only estimates $P(c|D^s)$ on the small $D^s$. The term $\mathcal{L}(c; \theta)$ is estimated over the large set as $\mathcal{L}(D^g \cap K(c); \theta)$ which uses many more samples and hence has less variance than the estimator $\mathcal{L}(D^s \cap K(c); \theta)$ over the small $D^s$.

We train CRISP models with stochastic optimization (Kingma & Ba, 2015, Adam) and propose Algorithm 1. Here, we do not explicitly reweigh the loss. We instead sample clusters from their importance. This avoids frequently visiting clusters with less weight. This strategy has less variance in its gradient estimates, which can help convergence (Seiffert et al., 2008; An et al., 2021). This algorithm is simple and efficient when one groups the generalist examples by cluster prior to training.

### 4 Experiments & Results

We perform experiments with transformer LMs (Vaswani et al., 2017). Most of our experiments use models with 1.3B parameters (trained on 120B tokens) and we conduct ablations with 350m and 7B models (resp. trained on 40B, 350B tokens). Our settings for architectures and optimization are borrowed from Brown et al. (2020), see Appendix D.

Our generalist training set is Redpj2 (Together AI Team, 2023). We select this dataset as it contains only web-crawled data without additional interventions to help evaluation tasks (e.g. adding

---

**Algorithm 1** CRISP Training

---

1: **Parameters:** $T$ (number of steps), $B$ (batch size)
2: **Input:** $D^s$ (specialist set), $D^g$ (generalist set)
3: $h^s \leftarrow \{P(c|D^s), \forall c\}$       ▷ Count cluster frequency on the specialist set $D^s$.
4: $\theta_0 \leftarrow \text{InitModel}()$            ▷ Initialize the model.
5: **for** $t = 1, \ldots, T$ **do**
6:   **for** $i = 1, \ldots, B$ **do**
7:     $c_i \sim \text{Categorical}(h^s)$    ▷ Sample a cluster id from the specialist histogram.
8:     $x_i \sim \text{Uniform}(D^g \cap K(c))$  ▷ Sample a generalist example in the selected cluster.
9:   **end for**
10:   $\theta_t \leftarrow \text{AdamUpdate}(\theta_{t-1}, \{x_1, \ldots, x_B\})$
11: **end for**

---

encyclopedias, books or academic articles). Redpj2 contains over 30T tokens with our 32k byte-pair encoding tokenizer (Sennrich et al., 2016b), see Table 4 in Appendix C. We segment the dataset into non-overlapping 1,024 token windows (the model context limit) and compute SBERT embedding for every window. We cluster the generalist dataset hierarchically with a clustering tree with branching 64 for 4 levels, see Appendix B. The levels therefore have 64, 4,096 ($= 64^2$), 260k ($= 64^3$) and 16.7m ($= 64^4$) clusters with an average of 540B, 8.4B, 130m and 2m tokens per cluster respectively. As an alternative to SBERT embeddings, we also consider Latent Semantic Index (LSI), i.e. singular value decomposition over tf-idf representations (Deerwester et al., 1990; Dumais, 2004).

For our specialist tasks, we consider 3 language modeling tasks (LM) and 3 multiple-choice-question tasks (MCQ). For LM, we use Pile subsets from different domains (Gao et al., 2021): medical (Pubmed Central), programming Q&A (Stackexchange), and encyclopedic (Wikipedia). For MCQ answering, we use AI2 Reasoning Challenge (Clark et al., 2018, ARC), Massive Multitask Language Understanding (Hendrycks et al., 2021, MMLU), and Reward Bench Reasoning (Lambert et al., 2024, RWDB-R). ARC focuses on science questions, MMLU focuses on interdisciplinary knowledge, RWDB-R focuses on correct vs incorrect solutions to math and programming problems. To provide a representative specialist train set $D^s \sim \mathcal{D}^s$, we split the questions into a train and test split, see Table 5 in Appendix C.

Our main results are reported with unified settings. For the classifier, the classification threshold is the main parameter. A threshold accepting 2.5% of $D^g$ worked best in for the runs with 1.3B models over 120B tokens. For DoGE, the method is costly to apply over many data sources/clusters and we applied it over 64 clusters, i.e. learning a mixture weight of dimension 64. For importance sampling, the results presented in this section relies on 260k clusters. Later, Section 5 studies ablations and parameter sensitivity. Details on hyperparameters can be found in Appendix D.

## 4.1 LANGUAGE MODELING TASKS

We evaluate specialist LMs on three domains from the Pile (Gao et al., 2021): medical (PubMed Central), encyclopedic (Wikipedia) and programming Q&A (StackExchange). We limit specialist training data from 14m tokens to the full Pile subset, up to 26.7B tokens, see Table 4 in Appendix C.

As baselines, we either train only on the in-domain (specialist) data without pretraining or we fine-tune a model pre-trained on Redpj2. We refer to the Redpj2 pretraining distribution as the base distribution. For task-dependent pretraining, we resample the Redpj2 pretraining set using a classifier, DoGE or importance sampling for each domain. The three methods have access to 14m specialist training tokens. In each case, the resampled pretraining set is used to train a 1.3B-parameter transformer model with the same hyperparameters as the Redpj2 baseline.

We report pretraining results in Figure 2, and the fine-tuning results in Figure 3. For each domain, the pretraining results evaluate models trained using the resampled Redpj2 examples. The fine tuning results evaluate models where each model pretrained on (resampled) Redpj2 has been further trained on the in-domain data itself (PubMed, StackExchange, Wikipedia). All experiments consider the same optimization effort and we validate the fraction of steps spent in fine-tuning, from 3%-ft with 14m tokens (97% pretraining) to 100%-ft with 26.7B tokens (no pretraining).

The pretraining results in Figure 2 show that the in-domain perplexity is better with task-dependent pretraining than with generic pretraining (base Redpj2) for all methods. This gain in perplexity comes as model training focuses on data close to the targeted domain: the model capacity is not used to fit the filtered out training data. Table 10 in Appendix F shows, for instance, that CRISP outperforms base on 97.3% of PubMed but reports worse perplexity on 95.9% of Redpj2.

When we fine tune the pretrained models, the advantage of task-dependent pretraining is preserved, as shown in Figure 3. Task-specific pretraining checkpoints are better starting points for fine-tuning than generic ones. This shows the complementarity between task-dependent pretraining and fine-tuning. Figure 3 also shows the necessity of pretraining: below 7B tokens, the "only specific" 1.3B model shows high perplexity. When comparing task-dependent pretraining methods, importance sampling consistently performs better after fine-tuning, even when the pretraining results are close (e.g. classifier on PubMed, Wikipedia).

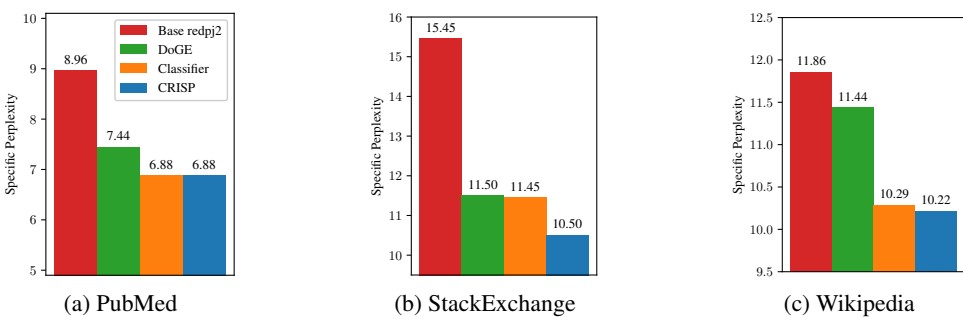

Figure 2: **Pretraining perplexities for language modeling tasks**

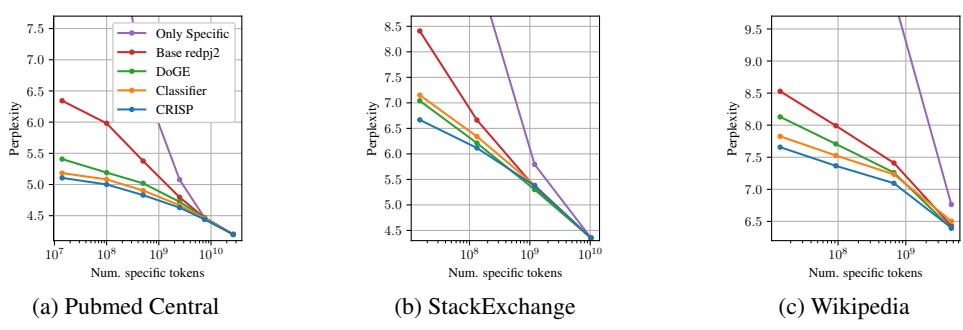

Figure 3: **Fine-tuned perplexities for language modeling tasks.** Task-dependent pretraining is always better than generic pretraining. The ordering of the methods is unchanged from pretraining.

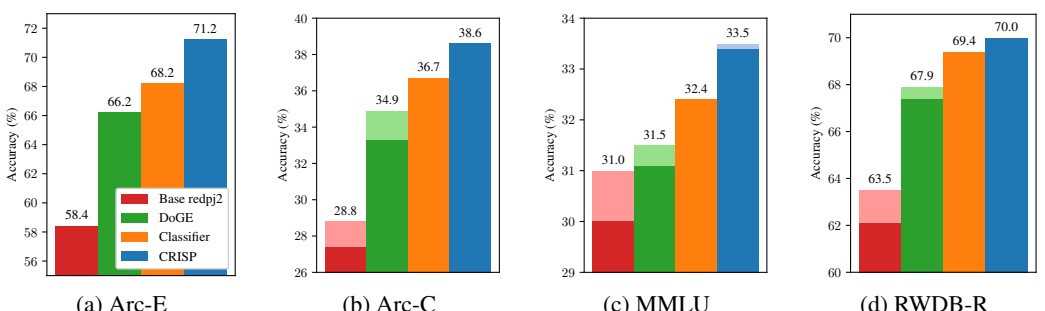

Figure 4: **Accuracy for multiple choice question tasks.** Light colors indicate fine tuning improvements if any. The ordering of the methods is consistent across all 4 datasets.

## 4.2 MULTIPLE CHOICE QUESTION TASKS

Compared to LM, MCQ has much smaller specialist training sets per task, i.e. between 200k and 2m tokens, see Table 5 in Appendix C. The MCQ evaluation is also different: it uses accuracy and not perplexity. For each example, the model scores the concatenation of the question and a possible answer, for each proposed answer. The model is accurate when the correct answer is assigned the highest score (probability or normalized probability, see Appendix E). For MCQ tasks, unlike for LM tasks, the training loss (negative log likelihood) is therefore not closely tied to the test metric.

Despite these differences, we observe a similar benefit for task-dependent pretraining compared to task-agnostic (base) pretraining. Figure 4 displays a similar method ordering and CRISP is consistently the best method. As a difference with LM tasks, we observe limited benefits from fine tuning, see Figure 4. Fine-tuning improves the base method on all datasets except ARC-E, but not enough to outperform task-specific pretraining, see Table 12 in Appendix G.

## 5 ANALYSIS

### 5.1 CLUSTERING

We study the impact of the text representation for clustering and the number of clusters. We consider two representations for clustering, the SBERT embeddings used in all other experiments and LSI embeddings, see Section 4. We report their performance with 64, 4096, 262k and 16.7m clusters.

The representation is important: examples in the same cluster are close in the embedding space. Our independence assumption, Equation 1, assumes that the loss in a cluster $c$ is the same regardless whether its data originates from $D^g$ or $D^s$, i.e.

$$\mathcal{L}(D^g \cap K(c); \theta) \simeq \mathcal{L}(D^s \cap K(c); \theta). \tag{3}$$

In practice, it is sufficient that the embedding space reflects the similarity of the loss gradient, i.e. if the gradients of the loss over a generalist cluster $D^g \cap K(c)$ is correlated with the gradient over a specialist cluster $D^s \cap K(c); \theta)$, the model trained on the former improves on the later. Figure 5 shows that the SBERT representation yields better results than LSI for all settings.

The number of clusters is a trade-off, and its optimum is at 260k for most of our experiments. There are multiple factors at play when the number of clusters varies. A smaller number of clusters implies larger clusters: our hypothesis, Equation 3, is then stronger, as it assumes loss similarity on large areas of the embedding space. At the limit, with one cluster, this hypothesis assumes that the specialist loss and generalist loss are identical everywhere. Conversely, as the number of clusters gets larger, the estimation of the cluster density on the small specialist set $P(c|\mathcal{D}^s) \simeq P(c|D^s)$ gets less accurate. The estimator risks overfitting, i.e. favoring clusters frequent in the training set $D^s$ but not as frequent on other samples from $\mathcal{D}^s$. Increasing the number of clusters also risks reducing the effective training set size: the specialist data could be mostly concentrated in a few clusters, corresponding to a small fraction of the overall generalist set $D^g$.

We explore these aspects on MMLU. We first measure the number of repeated examples when training models with CRISP pretraining for different number of clusters. Figure 6 shows the number of repetitions for each quantile of the training set. Even for 16.7m clusters, only a small minority of training examples are repeated beyond 10 times and the average number of occurrences of the training is examples 1.95, well within commonly recommended values (Muennighoff et al., 2023a; Xue et al., 2023).

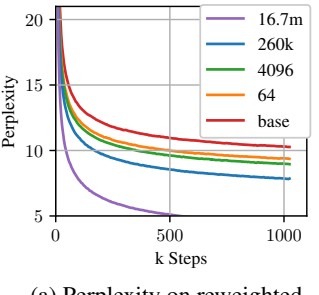
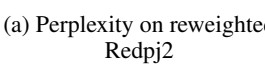

(a) Perplexity on reweighted Redpj2

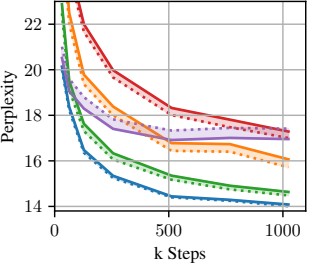

(b) Perplexity on MMLU train (plain) and test (dotted) sets.

Figure 8: **Perplexity for CRISP on MMLU with different number of clusters.** Y-scales on (a) and (b) are different.

Even if exact repetitions do not account for the poorer performance of the 16.7m cluster setting, its

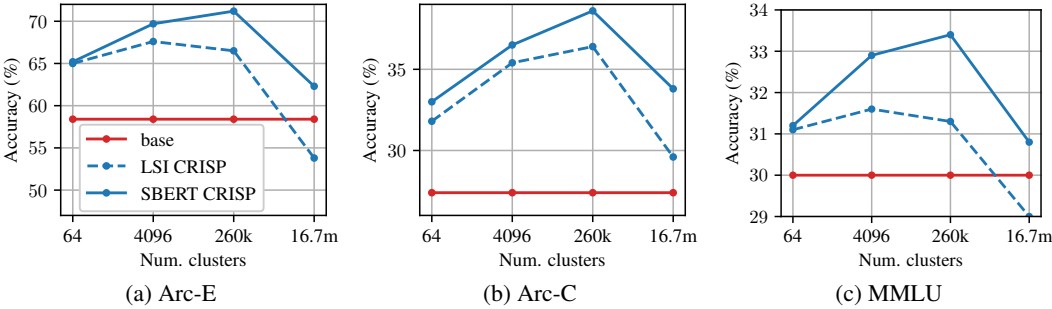

(a) Arc-E        (b) Arc-C        (c) MMLU

Figure 5: **Accuracy for multiple choice question tasks varying the text representation for clustering and the number of clusters.** SBERT is more effective than LSI in all cases.

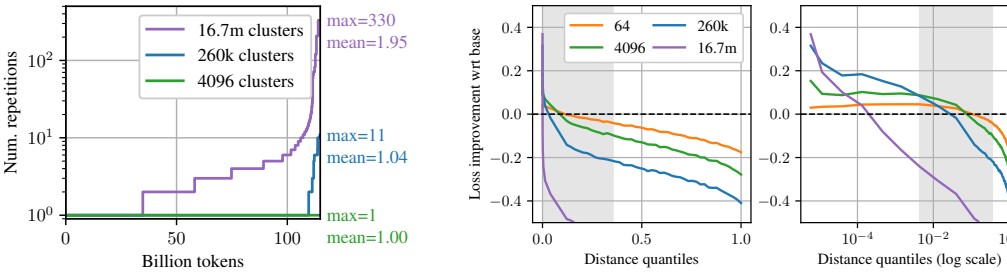

Figure 6: **Number of occurrences of each training example for CRISP on MMLU.** Repeated examples increase with the number of clusters.

Figure 7: **Loss improvement on Redpj2 (valid) wrt base as a function of the SBERT distance to MMLU train.** Models with a large number of clusters are better than base in a small area near MMLU train. The gray area indicates the 25-75% quantiles for the MMLU test set.

training set might be less diverse and the model might generalize well only in a small neighborhood of its training set. We evaluate if the Redpj2 examples with good perplexity concentrate around $D^s$, the MMLU training set. Figure 7 shows that the benefit of CRISP over base is indeed correlated with the distance to $D^s$. As the number of cluster increases to 16.7m, the benefit over base concentrate in an area with very few samples. For comparison, we plot the 2 middle quartiles [0.25, 0.75] where most of the MMLU test data concentrate in gray. We remark that MMLU test data mostly lies in an area where the perplexity of IS 16.7m is low.

Figure 8 shows the perplexity for CRISP runs on MMLU. On Figure 8a, the perplexity is computed from the reweighed loss on Redpj2. This is the loss optimized during pretraining. It shows that when the number of cluster increases the sampled training set is less diverse and corresponds to an easier learning problem ($< 5$ PPL). On Figure 8b, the perplexity is computed on the MMLU data itself, on the training set (plain) and on the test set (dotted). The scale of both plot is different: the resampled perplexities on Redpj2 are therefore not a good approximation of the MMLU perplexities. This quantifies the error resulting from our assumption, Equation 3. We also see overfitting for 16.7m clusters, the only case with better MMLU perplexity for train than for test. Finally, we notice that the gray area in Figure 7 fails to show that 260k cluster would have the best perplexity, which shows that SBERT distance to the training data is not the only factor explaining model performance.

## 5.2 MODEL SIZE

This section compares CRISP and base at 3 model sizes. The benefit of task-dependent training is consistent across model sizes, see Figure 9. We consolidate results across sizes to report the training cost in GPUh vs accuracy in Figure 10. GPUh are measured in training hours per graphic processor (Nvidia H100). We evaluate multiple checkpoints across model sizes and sort the checkpoints by training cost. The big dots mark transitions between model sizes: they show that the the 1.3B I.S. model outperforms the 6.7B base model on ARC. This shows substantial training speedups ($\sim$30x). Of course, a smaller model is also beneficial at inference.

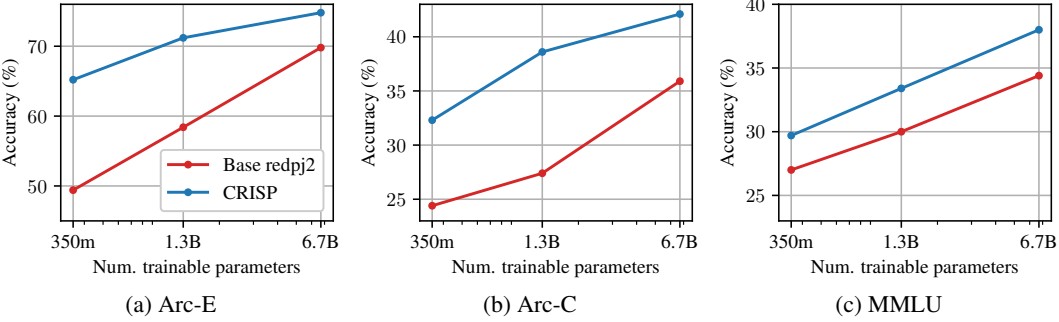

Figure 9: **Accuracy for multiple choice question tasks across model sizes.**

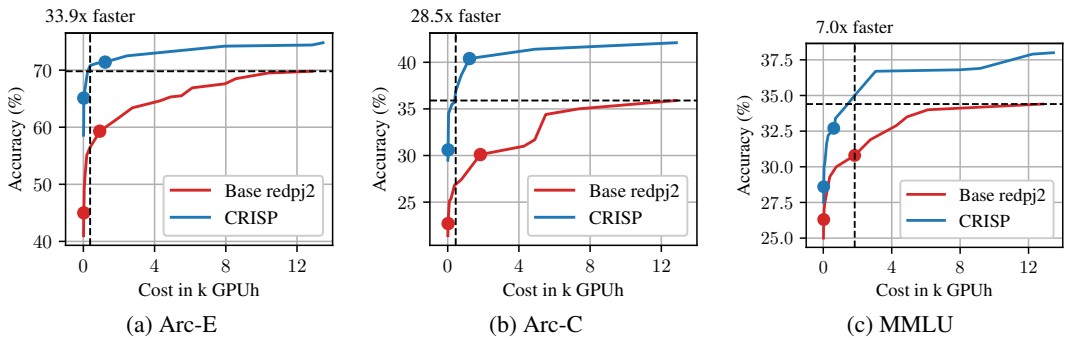

Figure 10: **Accuracy for multiple choice question tasks as a function of training cost.** The large dots mark the transition between model sizes (350m → 1.3B → 6.7B).

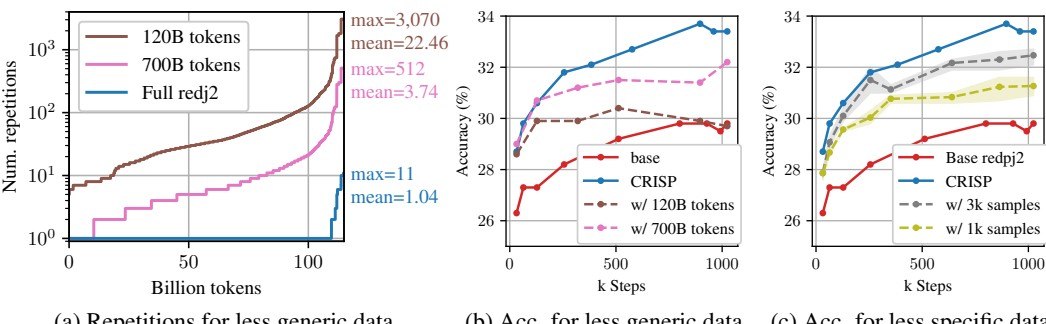

Figure 11: **MMLU with less training data.** When the generalist set $D^g$ is small (a,b), the importance sampling method will up-sample a small part of $D^g$ and this part will be seen multiple times during training. When this part is too small, the benefit of data selection vanishes. When the specialist set $D^s$ is small (c), the importance sampling weights are poorly estimated and the importance sampled data might not be representative of the targeted task.

## 5.3 DIFFERENT AMOUNT OF TRAINING DATA

This section varies both the amount of generalist data available to sample the CRISP dataset from and the amount of specialist data for inferring the CRISP weights. When specialist data concentrates on a few clusters, CRISP often samples generalist data from the same clusters, which can be problematic when the generalist set is small. We restrict the pretraining set to 700B and 120B tokens (downsampling Redpj2 by resp. $\sim$ 50x and $\sim$300x). Our pretraining runs use 120B tokens, so a base run never repeats in all settings. When CRISP is applied, some tokens are repeated. Figure 11a shows that, when restricting to 120B tokens, the number of repetition becomes high (22.5 on average) and CRISP is ineffective after 256k steps.

Table 1: **Accuracy (%) for Task Transfer and Multitasking.** Importance Sampling on MMLU and on multitask improves all tasks compared to baseline.

| Model | | Evaluation Tasks | | | | Multi |
|-------|--|------|------|------|--------|-------|
| | | ARC-E | ARC-C | MMLU | RWDB-R | |
| Base | Redpj2 | 58.4 | 27.5 | 30.1 | 62.2 | 45.1 |
| CRISP | ARC | **71.3** | **38.6** | 28.9 | 60.9 | 48.2 |
| | MMLU | 63.4 | 28.7 | **33.4** | 65.2 | 48.2 |
| | RWDB-R | 42.4 | 23.4 | 26.4 | 70.1 | 43.1 |
| CRISP | Multi | 68.6 | 34.1 | 31.1 | **70.9** | **51.1** |

When the specialist dataset is smaller, Figure 11c shows that the errors in estimating cluster frequencies $P(c|\mathcal{D}^s)$ negatively impact end task accuracy. This suggests future work to improve this estimation for tasks with small $D^s$: e.g. specific set augmentations or task grouping.

## 5.4 TASK-TRANSFER AND MULTITASKING

We perform cross-task evaluation, i.e. targeting a task A and evaluating on a task B, we also pretrain a multitask models with CRISP averaged weights from multiple tasks. Our results for the 1.3B models are in Table 1, we also report cross-task evaluation results for different model sizes in Appendix J. Cross-task evaluations show that, perhaps unsurprisingly, the best results on a task A are obtained when pretraining for task A. Transfer differs across tasks: CRISP targeting MMLU gives better results than base for all tasks, which is not the case for CRISP targeting ARC or RWDB-R. The multi-task result which mixes the histograms with the same weight (1/3 for ARC, MMLU and RWDB-R) gives the best result on averaged multitask accuracy. Surprisingly, on RWDB-R, this setting slightly outperforms targeting RWDB-R itself.

## 5.5 TASK-DEPENDENT CONTINUED PRETRAINING

We have seen the benefit of pretraining a model per task with CRISP in Figure 4. For tasks where pretraining cost is a concern, shorter pretraining runs still provide benefits, see Figure 10. Pretraining a multi-task model is also a cost-effective option, see Table 1. This section evaluates a third cost-effective option when targeting multiple tasks: continued pretraining. In this case, pretraining is divided into a generic pretraining phase and a task-dependent continued pretraining phase using CRISP. The compute cost of the generic pretraining can be shared across multiple tasks. Our results in Figure 12 show that even 10% of CRISP continued pretraining (i.e. generic pretraining for 928 steps out of 1,024) gives an

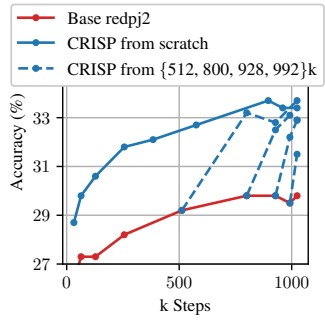

Figure 12: Continued Pretraining on MMLU

accuracy (32.9%) close to a full CRISP run (33.4%). We also remark that the impact of continued pretraining is stronger than fine tuning a generic model on MMLU (31.0% accuracy), see Figure 4.

## 6 CONCLUSIONS

A small specialist LM is interesting since it can outperform a larger generalist LM on its targeted domain while having a lower inference cost. We explore pretraining specialist LMs when little specialization data is available, a common setting that prevents pretraining of dedicated LMs. We evaluate different methods that modify the distribution of a generic training set guided by little specialist data. Our experiments highlight the benefit of clustered importance sampling: i.e. resampling the generic set such that its cluster histogram matches the specialist data. Our findings show that pretraining with this method provides strong models both for LM and question answering tasks. We also explore ways to lower the training cost of specialist models by showing their benefit on shorter training runs, continued pretraining and multitask settings. Our work shows that a simple, scalable importance sampling method can provide effective specialist LMs, even from little specialization data. Since clustered importance sampling is modality-agnostic, we foresee extensions of this work to other modalities, including vision and audio.

ACKNOWLEDGMENTS

We thank Angelos Katharopoulos, Matteo Pagliardini and Anastasiia Filippova for their advice throughout this project. We thank the anonymous reviewers for their suggestions and comments.

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

APPENDIX

## A  SCALABLE CLUSTERING

We cluster the generic dataset (Redpj2) with hierarchical clustering. We build a clustering tree. Each node in the tree is associated with a cluster centroid. The examples traverse the tree from top to bottom, selecting the node corresponding to the closest centroids among the current node's children.

The training of the tree proceed from root to leaves. Iteratively, a new level is built by applying k-means to a subset of the examples belonging to each node. We built a tree of depth up to 4, always splitting nodes in 64 clusters. For k-means, we normalize the Euclidean norm of the vectors prior to clustering. We train the model via Expectation Maximization using k-means++ initialization (Arthur & Vassilvitskii, 2006). At each step, we sample 6,400 new examples. With 20 steps, we visit 128k examples. To ensure a cluster distribution close to uniform, we monitor the cluster sizes at each assignment steps. If a cluster is larger than our balancing limit ($0.022 \simeq 1.5 * 1/64$), we split evenly at random its assignments with the smallest cluster, as suggested by Jegou et al. (2010). The clustering hyper-parameters can be found in Table 8.

## B  CLUSTERING & EMBEDDING COST

The computational cost of k-means clustering is negligible compared to the cost of computing text embeddings. Most of the experiments in this work are performed with SBERT `MiniLM-L6-v2` Reimers & Gurevych (2019). Embedding the 34.6T tokens of Redpj2 amounts to 5.4k GPU hours on a reference NVidia H100 GPU. Other models can provide better embeddings at a higher cost. We examined the results of clustering accuracy, MTEB evaluation Muennighoff et al. (2023b), versus embedding cost (for Redpj2 in GPU hours on H100) in Table 2.

Table 2: Embedding Cost versus Clustering Accuracy.

| Clustering method | | Cost (GPUh) | Accuracy (%) |
|---|---|---|---|
| `all-MiniLM-L6-v2` | Reimers & Gurevych (2019) | 5.4k | 41.94 |
| `e5-large-v2` | Wang et al. (2022) | 91.4k | 44.26 |
| `e5-base-v2` | Wang et al. (2022) | 27.4k | 44.10 |
| `all-mpnet-base-v2` | Reimers & Gurevych (2019) | 28.6k | 43.69 |
| `gte-base-v1.5` | Li et al. (2023) | 87.4k | 47.90 |
| `gte-small` | Li et al. (2023) | 13.3k | 44.89 |

In this cost-benefit table, `gte-small` stands out. We clustered the Redpj2 dataset with embedding from this model and we report MCQ results with CRIPS over this clustering. We compare these results with LSI and SBERT from Figure 5 in the main text. The results in Table 3 show that these embeddings are beneficial, especially with a small number of clusters.

## C  DATASET STATISTICS

Our generic pretraining set is Redpj2 (Together AI Team, 2023). We use the head+middle English version of the dataset, i.e. web-documents with a high density of English text. Our specialization datasets for language modeling are much smaller, see Table 4. Compared the LM tasks, the multiple choice question tasks have even smaller specialization training set, i.e. between 200k and 2m tokens, see Table 5. For the LM data, we rely on the train split provided by Pile Gao et al. (2021). For the MCQ data, we split each evaluation set into an equal sized train and test set uniformly at random. This provides a representative specialist train set $D^s \sim \mathcal{D}^s$. This also avoids cross-contamination between tasks, e.g. the official training set of MMLU contains ARC which would prevent the task transfer experiments in Section 5.4.

## D  ARCHITECTURES & HYPERPARAMETERS

Our architecture configurations are borrowed from Brown et al. (2020) and described in Table 6. We report the data selection hyperparameters in Table 7 and the clustering hyper-parameters in Table 8.

Table 3: MCQ accuracy with CRISP for different embedding methods.

| Clusters | Emb. | Arc-E | Arc-C | MMLU |
|---|---|---|---|---|
| 64 | LSI | 65.0 | 31.8 | 31.1 |
| | SBERT | 65.2 | 33.0 | 31.2 |
| | GTE | **67.8** | **33.8** | **31.5** |
| 4096 | LSI | 67.6 | 35.4 | 31.6 |
| | SBERT | 69.7 | 36.5 | 32.6 |
| | GTE | **69.9** | **37.0** | **32.8** |
| 262k | LSI | 66.5 | 36.4 | 31.3 |
| | SBERT | **71.2** | **38.6** | **33.4** |
| | GTE | 69.3 | 37.6 | **33.4** |
| 16m | LSI | 53.8 | 29.6 | 29.0 |
| | SBERT | 62.3 | 33.8 | 30.8 |
| | GTE | N/A | N/A | **31.7** |

Table 4: LM Datasets.

| | | Redpj2 | PubMed | StackExchange | Wikipedia |
|---|---|---|---|---|---|
| Dataset role | | *generalist* | *specialist* | *specialist* | *specialist* |
| Train | Num. tokens | 34.6T | 26.7B | 10.3B | 4.68B |
| | Num. documents | 24.0B | 2.94m | 15.4m | 5.79m |
| Test | Num. tokens | 359m | 52.4m | 20.1m | 14.1m |
| | Num. documents | 248k | 5.82k | 29.9k | 17.4k |

Table 5: MCQ Datasets.

| | | ARC-E | ARC-C | MMLU | RWDB-R |
|---|---|---|---|---|---|
| Train | Num. tokens | 143k | 79.6k | 2.05m | 426k |
| | Num. questions | 1.18k | 578 | 6.95k | 736 |
| | Avg. tokens per choice | 30.3 | 34.5 | 73.5 | 289 |
| Test | Num. tokens | 144k | 87.7k | 2.09m | 408k |
| | Num. questions | 1.19k | 593 | 7.09k | 695 |
| | Avg. tokens per choice | 30.2 | 37.0 | 73.6 | 293 |
| Num. choices per question | | 4 | 4 | 4 | 2 |

Table 6: Model Hyperparameters

| Num. parameters | 350m | 1.3m | 6.7B |
|---|---|---|---|
| Architecture | | | |
|   Embedding dim. | 1,024 | 2,048 | 4,096 |
|   Latent dim. | 4,096 | 8,192 | 16,384 |
|   Num. heads | 16 | 16 | 32 |
|   Depth | 24 | 24 | 32 |
|   Context limit | 1,024 | 1,024 | 1,024 |
| Optimization | | | |
|   Batch size | 96k | 115k | 1.04m |
|   Learning rate | 1e-4 | 1e-4 | 3e-4 |
|   Grad clipping | 5.0 | 5.0 | 0.1 |
|   Steps | 400k | 1m | 340k |
|   Num. train tokens | 40B | 120B | 350B |

# E  MCQ EVALUATION

For multiple-choice questions, we use the LM eval harness (Gao et al., 2024). For each task, the evaluated model estimates the (log) probability of each answer $a$ given the context $c$, that is, $\log P(a|c)$.

Table 7: Data-Selection Hyperparameters

| Method | Parameter | Range |
|---|---|---|
| Classifier | Regularization strength | {None, 1000, 100, 10, 1, 0.1, 0.01, 0.001} |
| | Threshold quantiles | {0.5, 0.6, 0.7, 0.75, 0.8, 0.9, 0.95, ... |
| | | ..., 0.975, 0.98, 0.9875, 0.99, 0.995, 0.9975} |
| DoGE | Num. clusters | 64 |
| | Proxy model size | Transformer base, 110m parameters |
| | Proxy model optimization | 32k batch size, 1e-4 learning rate, 100k steps |
| | Bregman coefficient $\mu$ | 5e-4 |
| | Transferred weights | {run average, last 20 step average} |
| Importance S. | Num. clusters | {64, 4096, 262k, 16.7m} |

Table 8: Hierarchical Clustering Hyperparameters

| Parameter | Range |
|---|---|
| Tree depth | 4 |
| Tree arity | 64 |
| Balancing limit | 0.022 |
| Number of samples per step | 6,400 |
| Number of steps | 20 |
| SBERT model | `MiniLM-L6-v2` |
| SBERT emb. dim. | 384 |
| LSI dim. | 256 |

The question contains the task prompt concatenated with the current question, while the answer contains the answer text. With this strategy, the model has no access to alternative answer choices proposed in the prompt. Table 9 reports our prompt. For all evaluations, we use the above prompt without example questions, that is, a zero-shot evaluation (Brown et al., 2020). Accuracy is calculated by verifying whether the highest score is assigned to the correct answer. The scores correspond to log probabilities for ARC-E and RWDB-R, while ARC-C, MMLU uses normalized scores, i.e. log probabilities divided by the number of characters in the answer.

Table 9: Task prompts (non-bold) for the multiple-choice-question tasks.

AI2 Reasoning Challenge (ARC) Easy and Challenge
Question: <question>\n
Answer: **<answer>**

Massive Multitask Language Understanding (MMLU)
The following are multiple choice questions (with answers) about <topic>.\n
Question: <question>\n
Answer: **<answer>**

Rewardbench Reasoning (RWB-R)
Follow the instructions below.\n
Instructions: <question>\n
Answer: **<answer>**

# F    SUPPLEMENTARY RESULTS FOR LANGUAGE MODELING TASKS

We measure the fraction of examples where the pre-trained model is better than the base model. We measure this rate both on the held-out data from $\mathcal{D}^g$ (measured on the 360m tokens from Redpj2 valid) and on held-out data from $\mathcal{D}^g$ (measured on the full Pile validation set). The results in Table 10 show that the model trained with importance sampling improves perplexity on most specialist

Table 10: **Fraction of examples with lower perplexity with importance sampling than with base.** Compared to base, CRISP models specialize: they performs better on most specialist examples and worse on most generic examples.

|  | Generalist $\mathcal{D}^{\mathrm{g}}$ (Redpj2) | Specialist $\mathcal{D}^{\mathrm{s}}$ (Pile subset) |
|---|---|---|
| PubMed | 6.1% | 97.3% |
| StackExchange | 2.9% | 92.6% |
| Wikipedia | 12.4% | 86.7% |

documents (right column). Its training on the importance sampled distribution utilize model capacity mostly on data close to the domain of interest, this relieves the model from fitting well most of the generic data, and hence most generic documents have higher perplexity with CRISP (left column).

For completeness, we also report the perplexity numbers of Figure 3 in Table 11.

Table 11: **Perplexity on language modeling tasks after fine-tuning.** These tables reports the perplexity numbers from Figure 3.

(a) PubMed

| Specific tokens | 14m | 100m | 500m | 2.5B | 7.5B | 26.7B |
|---|---|---|---|---|---|---|
| Only Specific | 25.73 | 10.09 | 6.64 | 5.08 | 4.47 | 4.20 |
| Base redpj2 | 6.34 | 5.98 | 5.38 | 4.79 | 4.47 | 4.20 |
| DoGE | 5.41 | 5.19 | 5.02 | 4.72 | 4.47 | 4.20 |
| Classifier | 5.18 | 5.08 | 4.90 | 4.67 | 4.45 | 4.20 |
| CRISP | 5.11 | 5.00 | 4.83 | 4.63 | 4.44 | 4.20 |

(b) StackExchange

| Specific tokens | 15m | 133m | 1.2B | 10.3B |
|---|---|---|---|---|
| Only Specific | 23.93 | 9.60 | 5.79 | 4.35 |
| Base redpj2 | 8.41 | 6.66 | 5.35 | 4.35 |
| DoGE | 7.04 | 6.21 | 5.30 | 4.35 |
| Classifier | 7.15 | 6.34 | 5.39 | 4.35 |
| CRISP | 6.67 | 6.12 | 5.38 | 4.35 |

(c) Wikipedia

| Specific tokens | 14m | 93m | 668m | 4.7B |
|---|---|---|---|---|
| Only Specific | 57.13 | 18.22 | 9.97 | 6.76 |
| Base redpj2 | 8.53 | 7.99 | 7.41 | 6.43 |
| DoGE | 8.13 | 7.71 | 7.26 | 6.39 |
| Classifier | 7.83 | 7.53 | 7.23 | 6.50 |
| CRISP | 7.66 | 7.37 | 7.09 | 6.40 |

# G SUPPLEMENTARY RESULTS FOR MULTIPLE CHOICE QUESTIONS

Table 12 reports the MCQ results before and after fine-tuning, i.e the accuracy numbers from Figure 4. Fine-tuning on the small MCQ train sets optimizing log-likelihood does not always benefit end-task accuracy.

# H COMPARING THE RESULTS OF DoGE AND IMPORTANCE SAMPLING

We observe in Table 13 that the pretraining results of DoGE and importance sampling on 64 clusters are close. Both methods pretrain models by sampling the clustered generalist data according to the cluster weights. If both methods would infer the same cluster weights, their pretraining runs would

Table 12: **MCQ Accurary.** Fine tuning results are dashed when not improved from pretraining. This table reports the accuracy numbers from Figure 4.

|  | ARC-E | | ARC-C | | MMLU | | RWDB-R | |
|---|---|---|---|---|---|---|---|---|
|  | Pretr. | +ft | Pretr. | +ft | Pretr. | +ft | Pretr. | +ft |
| Base redpjv2 | 58.4 | – | 27.4 | 28.8 | 30.0 | 31.0 | 62.1 | 63.5 |
| DoGE | 66.2 | – | 33.3 | 34.9 | 31.0 | 31.5 | 67.4 | 67.9 |
| Classifier | 68.2 | – | 36.7 | – | 32.4 | – | 69.4 | – |
| CRISP | 71.2 | – | 38.6 | – | 33.4 | 33.5 | 70.0 | – |

Table 13: DoGE & CRISP on 64 Clusters

|  | LM PPL↓ | MCQ Acc (%) ↑ | | |
|---|---|---|---|---|
|  | PubMed | ARC-E | ARC-C | MMLU |
| DoGE | 7.44 | 66.2 | 33.3 | 31.0 |
| CRISP | 7.28 | 65.2 | 33.0 | 31.2 |

be identical. We therefore ask if the similar results are due to similar cluster weights. Figure 13 compares the cluster weights for both methods. The top clusters for both methods are similar, but their histograms are not identical. This shows that similar pretraining results can be obtained with different weights.

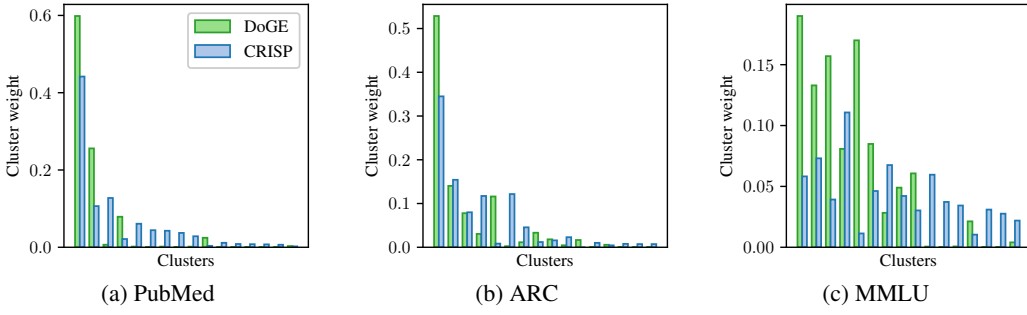

Figure 13: DoGE vs CRISP weights with 64 clusters. We report the top-16 clusters sorted by mean weight across methods.

## I COMPARING CRISP AND CROSS-ENTROPY DIFFERENCE (CED)

Contrasting the scores of two LMs (generalist and specialist) is a popular method for data selection (Moore & Lewis, 2010; Axelrod et al., 2011; Wang et al., 2018; Junczys-Dowmunt, 2018). We considered this method based on Wang et al. (2018): we obtain the specialist LM by fine-tuning a generalist LM on the specialist training set. We rely on a 350m parameter model for the selection. One should note that this method is particularly expensive since it requires scoring the entire Redpj2 dataset twice with an LM, which is more expensive than embedding and clustering the dataset. Our results show that CED improves over the classifier method but CRISP is significantly better, see Table 14.

## J TASK TRANSFER FOR 350M, 1.3B AND 7B MODELS

Table 15 complements the task-transfer results from Table 1 in Section 5.4 with the results across different model sizes. The importance sampling models trained with MMLU histograms outperform the base models on all tasks for all model sizes.

Table 14: **Comparison with Cross-Entropy Difference** for MCQ Accuracy (%), 1.3B model.

|            | Arc-E    | Arc-C    | MMLU     |
|------------|----------|----------|----------|
| Base       | 58.4     | 27.4     | 30.0     |
| CED        | 58.9     | 30.5     | 31.1     |
| Doge       | 66.2     | 33.3     | 31.0     |
| Classifier | 68.2     | 36.7     | 32.4     |
| CRISP      | **71.2** | **38.6** | **33.4** |

Table 15: Accuracy (%) for Task Transfer on 350m, 1B and 7B models.

| Model |            | Evaluation Tasks | | | | |
|-------|------------|-------|-------|------|--------|-------|
|       |            | ARC-E | ARC-C | MMLU | RWDB-R | Multi |
| 350m  | Base       | 49.5  | 24.5  | 27.0 | 57.6   | 40.5  |
|       | CRISP ARC  | 65.3  | 31.5  | 27.4 | 58.4   | 44.7  |
|       | CRISP MMLU | 55.6  | 26.3  | 29.8 | 61.3   | 44.0  |
| 1B    | Base       | 58.4  | 27.5  | 30.1 | 62.2   | 45.1  |
|       | CRISP ARC  | 71.3  | 38.6  | 28.9 | 60.9   | 48.2  |
|       | CRISP MMLU | 63.4  | 28.7  | 33.4 | 65.2   | 48.2  |
| 7B    | Base       | 69.9  | 35.9  | 34.4 | 64.9   | 50.7  |
|       | CRISP ARC  | 74.5  | 42.2  | 32.6 | 62.4   | 51.1  |
|       | CRISP MMLU | 70.0  | 37.6  | 38.0 | 67.5   | 53.1  |

