# OpenReview forum: "Task-Adaptive Pretrained Language Models via Clustered-Importance Sampling"
_ICLR.cc/2025/Conference — ICLR 2025 Poster_

### Official Review · Reviewer_TBtV · 2024-10-28

**Soundness:** 3
**Presentation:** 2
**Contribution:** 3
**Rating:** 6
**Confidence:** 4

**Summary:**

In this work, the authors investigate the data selection problem during the pre-training and continued pre-training stages of language models. They propose CRISP, a method that clusters both the generalist and specialist datasets, and then selects data from the generalist dataset based on the cluster distribution of the specialist dataset. The authors conduct extensive experiments on six benchmarks to demonstrate the effectiveness of CRISP.

**Strengths:**

1. The authors conduct comprehensive experiments, utilizing significant computational resources, to validate the effectiveness of the proposed CRISP method.
2. Although the computational cost of CRISP is not explicitly discussed, based on its methodology, CRISP appears to be computationally efficient.

**Weaknesses:**

1. The paper is somewhat hard to follow. For instance, CRISP requires a K-means clustering model to select task-specific data, but the authors do not mention on which dataset the K-means model is trained. Furthermore, many technical details are placed in the appendix, making the paper hard to follow and not self-contained.
2. CRISP essentially selects data from the generalist dataset based on the distribution of the specialist dataset. There are several baseline approaches that should be considered, such as Cross-Entropy Difference [1], as mentioned in the related work section.
3. Although the author conduct extensive experiments with substantial amount of compute, the scientific contributions of this work appear to be marginal and the findings are not particularly surprising. Like previous works, CRISP aims to align the selected data more closely with the targeted domain. It is unclear what unique strengths CRISP offers compared to prior methods. For example, given the vast amount of training data, it would be beneficial if the authors can demonstrate CRISP is significantly more efficient than existing approaches. Furthermore, the data quality in the selection process might be another factor to consider.


### Additional Comments (do not affect my rating)

1. Recent works [2, 3] have demonstrated that results on RewardBench have a negative correlation with real downstream performance. I suggest the authors consider using other benchmarks for evaluation. Although these findings are very recent and were reported after the paper submission deadline—and it's not the authors' fault—I believe it would be beneficial if they could adjust their evaluation settings accordingly.
2. The authors utilize significant computational resources for the experiments in this work. I appreciate these efforts. It would be helpful if the authors could present a comparison regarding the compute cost to better understand the pros and cons of the approaches discussed.

Reference:

[1] Moore, Robert C., and William Lewis. "Intelligent selection of language model training data." In Proceedings of the ACL 2010 conference short papers, pp. 220-224. 2010.

[2] Zhou, Enyu, Guodong Zheng, Binghai Wang, Zhiheng Xi, Shihan Dou, Rong Bao, Wei Shen et al. "RMB: Comprehensively Benchmarking Reward Models in LLM Alignment." arXiv preprint arXiv:2410.09893 (2024).

[3] Frick, Evan, Tianle Li, Connor Chen, Wei-Lin Chiang, Anastasios N. Angelopoulos, Jiantao Jiao, Banghua Zhu, Joseph E. Gonzalez, and Ion Stoica. "How to Evaluate Reward Models for RLHF." arXiv preprint arXiv:2410.14872 (2024).

**Questions:**

1. If I understand the proposed approach CRISP correctly, should line 8 of Algorithm 1 be $x_i \sim \text{Uniform}(D^{g} \cap K(c_i))$?
2. Typically, we use accuracy to measure performance on MMLU, while the authors compute perplexity on MMLU. How are the inputs formatted? Are the question and the corresponding correct option concatenated?
3. As shown in Algorithm 1 and Figure 1, CRISP resamples the generalist data based on the specialist data histogram. Is the specialist dataset not used in the training process?

---

> ### Author Response · Authors · 2024-11-21
> **Proposed changes to address TBtV's concerns**
>
> Thank you for reading our paper and for your comments. We can address the comments by adding the missing information in the text (W1, W3, Comment2, Q1, Q2, Q3). We can run additional experiments between now and the camera ready due date for W2 and Comment 1.
>
> **W1.** This is an important point that we omitted indeed. We will clarify in the text L152 and L158 that the clustering is trained on the generalist corpus D^g. Regarding the appendix, we are happy to move sections between the appendix and the main text, and would gladly hear your suggestions for such edits.
>
> **W2.** The literature on data selection methods is rich and we wanted to have a representative of each family of methods. [1] is actually very similar to DoGE (which is more recent) since it is actually related to gradient alignment (see sec 4.3 on Influence Functions in “The Trade-offs of Domain Adaptation for Neural Language Models”, Grangier and Iter, ACL 2022). We are currently running cross-entropy difference experiments. These experiments are not trivial in terms of cost since filtering involves scoring the full redpj2 corpus (34T tokens) with two LMs for each selection problem (while the other methods involve computing the sentence BERT embedding once for all selection problems).  If the experiments are done before the end of the discussion period we will provide the additional results below, in the camera ready otherwise.
>
> **W3.** We appreciate your comment and we might have gone quickly through some unique aspects of CRISP. We propose to stress two unique aspects of CRISP to address this concern, possibly in conclusions: (1) robustness to variation in the initial distribution (if some examples are repeated many times in the pre-training data, importance sampling naturally correct for that increased frequency which is not the case for most previous work based on thresholded classifiers), (2) CRISP clusters the generic data once for all tasks, only the histogram is computed per task (other methods run inference with a novel classifier or a gradient alignment for each task). We are also happy to cite and compare our conclusions with prior work that we did not consider yet, which references did you have in mind?
>
> **Comment1.** We were unaware of this recent result. Thank you for mentioning it. We are happy to take suggestions for a replacement dataset. Our goal is to have a multiple choice task for math and programming.
>
> **Comment2.** The cost of the classifier method and the cost of CRISP are similar. Both require computing the sentence BERT embedding for each document in the pretraining corpus.  The cost per token (for sentence BERT mini) is 280x smaller than the pre-training cost per token for a 1.3B-transformer model. The cost of training and inference for both the classifier and k-means are negligible compared to computing the sentence BERT representation. We will introduce an appendix section to report on the compute cost of inferring the sentence BERT embeddings,  pre-training/fine-tuning each model size. We will also report the cost of DoGE which is more costly as this method needs to pretrain a smaller model first.
>
> **Q1.**  Thank you for your careful reading. Indeed we need to correct D^s to D^g on line 8.
>
> **Q2.** We measure accuracy for zero-shot QA on MMLU. Table 1 and Figures 4, 5, 9, 10, 11 and 12 reports accuracy. Figure 8 reports the perplexity on MMLU with the concatenation provided in Appendix D, Table 7. We will add this information in the caption of Figure 8.
>
> **Q3.** Only the training histogram of the specialist dataset is used. We will clarify the sentence explaining our setup L235.  We report statistics for the training and evaluation fraction of each dataset in Appendix B, Table 3 and 4. We also have an ablation when the specialist training set is smaller in Sec 5.3, Figure 11c. Note that we can also use the specialist set for fine-tuning after pre-training using crisp, as shown in fig.3.

---

> > ### Comment · Reviewer_TBtV · 2024-11-22
> >
> > I thank the authors for their response. Most of my concerns are addressed and I will adjust my score accordingly.

---

### Official Review · Reviewer_Z9XR · 2024-10-31

**Soundness:** 2
**Presentation:** 3
**Contribution:** 2
**Rating:** 6
**Confidence:** 4

**Summary:**

This work focuses on training the specialist LM to avoid the limitation of scarce  specialist data.  To this end, this work explore three data selection method during pre-training in which the clustered importance sampling (named CRISP in this work) performs best. The CRISP aims to modifies the distribution of a generic training set guided by little specialist data. To be specific, the method resamples the generic set such that its cluster histogram matches the specialist data. Therefore, the model can always select the samples with the same or similar distribution with the targeted domains. Experimental results demonstrate that pre-training with this method provides better specialist models both for LM and question answering tasks. Besides, the additional analysis experiment benefits readers more, e.g., investigating the impact of model size, training set size, the number of clusters, clustering methods, continued pre-training and multi-task settings. All of the results show the advantage of the proposed method.

**Strengths:**

1. The authors extend the importance sampling (Grangier et al., 2024b) through adjusting the frequency of generalist clusters guided by specialist data. Besides, it scale the method to millions of clusters, show that it works with larger models, and extend it beyond language modeling tasks (i.e., adding the downstream tasks).
2. Extensive experiments show the importance of sampling data and the effectiveness of the cluster-based importance sampling method.
3. The cluster-based importance sampling method may benefit the researchers that devoted into the end-to-side small LLMs.

**Weaknesses:**

1. The cluster-based importance sampling method is not proposed by this work and the authors only show it works in larger models, and extend it to MQA tasks. Therefore, the contribution of technology is limited.
2. From my perspective, the CRISP method frequently selects samples (tokens) that share the same distribution as the target domain. What if you were to iterate over generalist samples/tokens—without distinguishing whether they belong to the same domain—the same number of times as you would when training specialist LMs?
3. When a new domain or downstream task coming, it has to be trained a new specialist LMs from scratch. It maybe time-cost and laborious.
4. If you apply the CRISP method based on a pre-trained generalist LMs, I wonder whether it can gain more benefit. If not, does it mean that you are still not using enough data in current pre-training setting?

**Questions:**

See the weaknesses.

---

> ### Author Response · Authors · 2024-11-21
> **Proposed changes to address Z9XR's concerns**
>
> Thank you for reading our paper and for your comments. We can improve the text on the points you raise in W1 and W3. We want to address W2 but we would appreciate clarifications from your side.
>
> **W1.** Our work effectively builds upon the method presented in the previous work that we cite. We want to modify the paper to stress the contributions compared to this earlier work. We indeed (1) perform experiments at a larger scale (all model sizes here 350m, 1.3B, 6.8B exceed the scale from [1], 126m) and (2) introduce hierarchical clustering for that. We perform experiments on (3) end-tasks (MCQ) while [1] is limited to LM. We also (4) explain the impact of hyperparameters such as the clustering representation and number of clusters. We studied (5) multitasking and (6) continued pretraining. We also perform (7) ablations pertaining to the generic pre-training set size and the specific data size. These contributions are (1, 2, 3, 4, 5, 6, 7) non-trivial and provide a complete study of the method to practitioners wanting guidance beyond the small scale experiments in [1]. We will add an explicit list of contributions at the end of the introduction.
>
> **W2.** Could you clarify W2? We want to stress that outside of the ablations in 5.3., sampling with the specific histogram barely yields any repeated tokens. Figure 6 shows the number of occurrences of the training tokens is 1.04 in the training run with the best setting (260k clusters), so less than 4% of tokens are repeated.
>
> **W3.** Indeed the pretraining cost could be prohibitive for some applications. We offer two solutions to limit this cost. The continued pre-training results (Sec 5.5) shows that, without drop of accuracy, we can limit task-specific pretraining to 20% of the pretraining, while the rest can be shared among tasks. The multitasking results (Sec 5.4) shows that improvements over the base run can be obtained by taking the model pre-trained on the distribution of another task (e.g. taking the MMLU model improves all tasks over base). We will stress these two points in the introduction.
>
> **W4.** We believe that Section 5.5 on continued pretraining evaluates exactly this setting: in that section, a LLM is first pre-trained on generic data with the standard distribution, yielding a pre-trained generalist LM. Then, after some number of iterations, we switch to CRISP, showing that CRISP is still beneficial as a continued pre-training method.
> We will add this sentence “We apply CRISP to a pre-trained generalist LM. We keep the overall number of training tokens fixed and explore different splits between the generic pretraining and the specialized CRISP pretraining.” Figure 12 reports settings with 50/50, 78/22, 91/9 or 97/3% of generic vs specific (CRISP) pretraining. CRISP is always beneficial and the full benefit of CRISP is preserved up to the 91/9% setting.

---

> > ### Comment · Reviewer_Z9XR · 2024-11-22
> >
> > Thanks for your response. Most of my concerns have been addressed and I will adjust my score.

---

### Official Review · Reviewer_oNof · 2024-11-03

**Soundness:** 4
**Presentation:** 4
**Contribution:** 3
**Rating:** 8
**Confidence:** 2

**Summary:**

This paper addresses the challenge of building specialist language models (LMs) when domain-specific data is limited. The authors introduce Clustered Importance Sampling (CRISP), which adapts a generalist dataset for specialist pretraining by adjusting its distribution to align with limited domain-specific data. CRISP’s effectiveness is demonstrated across various domains, yielding improvements in language modeling perplexity and multiple-choice question accuracy. The approach is scalable, applicable to both small and large models, reduces the need for costly domain-specific data, and consistently outperforms other adaptation techniques, such as classifiers and gradient alignment, across a range of tasks and domains.

**Strengths:**

S1. The problem is clearly defined and well-motivated, addressing the need for effective specialist language models when domain-specific data is limited.

S2. CRISP demonstrates superior performance compared to both untrained and pre-trained models on the Redpj2 baseline, as well as classifier-based and DoGE approaches. It achieves lower perplexity on language modeling tasks and higher accuracy on multiple-choice question (MCQ) tasks across four different domains.

S3. The paper provides a comprehensive analysis, exploring the effects of cluster numbers, model sizes, varying amounts of training data, task transferability, and multitask training. CRISP has also been shown to be more efficient in terms of training cost and scales well with larger models.

**Weaknesses:**

W1. While the paper provides a comprehensive analysis of different cluster numbers, it is limited in terms of guidance on determining the optimal number of clusters beyond empirical trial and error, even though this is a crucial parameter for CRISP’s performance. Results indicate that the number of clusters influences the sampling distribution and domain adaptation effectiveness, concerning token repetition or overfitting in certain configurations.

W2. Although CRISP improves model performance in non-pretrained scenarios, it offers limited additional benefits for MCQ tasks after fine-tuning compared to task-specific pretraining.

W3. When the specialist dataset $\mathcal{D}^s$ is very small, CRISP appears to be less effective in enhancing end-task accuracy.

**Questions:**

I have the following additional questions and comments:

- As shown between SBERT and LSI experiments, the choice of clustering model impacts performance. Would using a more powerful embedding model like E5-large instead of the current MiniLM-L6-v2 improve performance, considering factors such as computational cost (as using E5-large would be more costly) and potential performance gains?
-  How long does this process typically take to build the clustering tree for different configurations? I think Figure 10 only accounts for the training cost and it would be nice to compare it with the overall training cost.

I have also identified some potential typos:
- Currently, the equation on page 4 after equation (2) gives $\mathcal{L}(\mathcal{D}^{s}; \theta) = \sum\limits_{c} \mathcal{L}(c; \theta) \frac{P(c | \mathcal{D}^{s})}{P(c | \mathcal{D}^{g})} P(c | \mathcal{D}^{g}) = \underset{c \sim (c | \mathcal{D}^{s})}{\mathbb{E}} [w(c) \mathcal{L}(c; \theta)]$. Shouldn't this be $\mathcal{L}(\mathcal{D}^{s}; \theta) = \sum\limits_{c} \mathcal{L}(c; \theta) \frac{P(c | \mathcal{D}^{s})}{P(c | \mathcal{D}^{g})} P(c | \mathcal{D}^{g}) = \underset{c \sim (c | \mathcal{D}^{g})}{\mathbb{E}} [w(c) \mathcal{L}(c; \theta)]$, that is $c \sim (c | \mathcal{D}^{g})$ instead of $c \sim (c | \mathcal{D}^{s})$? In addition, if correction is needed, would this change the interpretation of the Importance Sampling as the equation changes?
- In Algorithm 1, I think line 8 should be $x_i \sim \text{Uniform}(D^{g} \cap K(c_i))$ as it samples a generalist example over a samples cluster id in the large set as explained in previous paragraphs. In addition, in line 10, the set should start with $x_1$ instead of $x_0$ to be consistent with the indexing.

---

> ### Author Response · Authors · 2024-11-21
> **Proposed changes to address oNof's concerns**
>
> Thank you for your attentive reading and comments. Regarding the weaknesses and questions, you are making good points that we want to address in the text.
>
> **W1**. The number of clusters should indeed be validated empirically. We will mention two points regarding the cost of this selection. (1) we will include in the appendix that the best number of clusters for MCQ accuracy after a long pretraining run is actually the same as the one determined by measuring the perplexity over validation data at early pre-training checkpoints. (2) the hierarchical clustering can be built once for all tasks, and then it can be reused by different tasks at different levels to select their number of clusters (we will edit the text to stress this advantage in Sec 5.1).
>
> **W2. and W3.** We will be more explicit about the limitations of this work when we mention our future work in conclusions. For W2, we will propose to evaluate the benefit of fine-tuning with an alignment loss (e.g. DPO), the objective of which is closer to the accuracy metric of MCQ. For W3, augmentations over the specific set (e.g. estimating cluster ids from multiple windows of text per specific example) could improve the estimation of the cluster histograms.
>
> **Q1.** This is a good suggestion. It would indeed be good to see if alternative neural representation improves the results in Figure 5. We looked at different models with different inference cost in (numbers of weeks on 1GPU H100 for redpj2) and clustering accuracy (on MTEB https://arxiv.org/abs/2210.07316 as reported on https://huggingface.co/spaces/mteb/leaderboard). e5-large-v2 is too costly for us but we are currently running gte-small which seems to be competitive in terms of accuracy. If these experiments are done before the end of the discussion period we will provide the additional results below, in the camera ready otherwise.  We hope that this addresses your concerns.
>
> |Model             |Cost     |Acc       |
> |------------------|---------|----------|
> |all-MiniLM-L6-v2   |       32|     41.94|
> |e5-large-v2        |      544|     44.26|
> |e5-base-v2        |      163|     44.10|
> |all-mpnet-base-v2 |      170|     43.69|
> |gte-base-v1.5     |      520|     47.90|
> |gte-small         |       79|     44.89|
>
> **Q2.** The cost of clustering is limited compared to LM pretraining. The cost of clustering comes from two steps: (1) computing the sentence BERT representation for each document in the pretraining corpus, (2) training and applying the hierarchical clustering on these representations. This cost of (2) is negligible compared to (1). The cost of (1) per token (for sentence BERT mini) is 280x smaller than the training cost per token for a 1.3B-transformer model. We will introduce an appendix section to report on the compute cost of inferring the sentence BERT embeddings,  pre-training/fine-tuning each model size.
>
> **Typos:** Thank you for your careful reading. We will correct the 3 mentioned typos.
> - The last expectation is indeed over c|D^g.
> - The samples from line 8 are from D^g. And line 10 should start at x_1 not x_0.

---

> > ### Comment · Reviewer_oNof · 2024-11-22
> >
> > Thank you for your response and the adjustments you plan to make! I believe most of my concerns and questions have been addressed, therefore I will keep my original score.

---

### Official Review · Reviewer_RQp5 · 2024-11-08

**Soundness:** 3
**Presentation:** 3
**Contribution:** 2
**Rating:** 6
**Confidence:** 2

**Summary:**

This work proposes a novel method to pre-train a specialist language model, where the amount of specialist data is scarce, and the amount of generalist data is rich. The high-level idea in this method is to sample from the generalist data and select the ones that are more similar to the specialist data. The authors target it with a simple cluster-then-importance-sampling, showing improvement in perplexity and accuracy on language model and multiple choice QA tasks.

**Strengths:**

* The proposed method is simple while effective, showing competitive performance compared to vanilla pre-training and other baselines.

* The ablation study in Section 5 is comprehensive and well-explored.

**Weaknesses:**

* Previous work [1] has proposed a similar method of adjusting generalist data according to clusters informed by specialist data. The major contribution of the method is that this work shows that that method will work when scaled up.

[1] Specialized Language Models with Cheap Inference from Limited Domain Data

**Questions:**

It would be helpful if the authors could provide a more detailed association with previous work.

---

> ### Author Response · Authors · 2024-11-21
> **Proposed changes to address RQp5's concerns.**
>
> Thank you for your review. We are glad that you mentioned the simplicity and effectiveness of our method and that you found our ablations thorough.
>
> Our work effectively builds upon the method presented in the previous work that we cite. We want to modify the paper to stress the contributions compared to this earlier work. We indeed (1) perform experiments at a larger scale (all model sizes here, 350m, 1.3B, 6.8B exceed the scale from [1], 126m) and (2) introduce hierarchical clustering for that. We  (3) perform experiments on end-tasks (MCQ) while [1] is limited to LM. We also (4) explain the impact of hyperparameters such as the clustering representation and number of clusters. We studied (5) multitasking and (6) continued pretraining. We also perform ablations pertaining  (7) for the generic pre-training set size and the specific data size. These contributions (1, 2, 3, 4, 5, 6, 7) are, from our point of view, non-trivial and provide a complete study of the method to practitioners wanting guidance beyond the small-scale experiments in [1]. We will add an explicit list of contributions at the end of the introduction.

---

> > ### Comment · Reviewer_RQp5 · 2024-12-02
> >
> > Thanks for your clarification. It addresses my concern and I will keep my score positive.

---

### Author Response · Authors · 2024-11-21
**Summary of the proposed changes to address the reviewers' concerns.**

We thank the reviewers for their careful reading of our paper and for their questions which help us improve the paper. We appreciate that all reviewers found the experimental validation of CRISP convincing.

We summarize the main actions we want to take to address their comments:
1. Highlight our contributions compared to (Grangier et al, arxiv 2402.01093), in answer to RQp5, Z9XR.
2. Clarify that the clustering is learned on the generic corpus [TBtV]
3. Evaluate another neural representation for clustering as an alternative to sentence BERT  [oNoF].
4. Report the computational cost for clustering, the inference of sentence BERT representations and the classifier baseline [oNoF, TBtV].
5. Include Cross-Entropy difference as a baseline [TBtV]
6. Remove results on RWD-bench/math and possibly find an alternative evaluation set for math and coding [TBtV].

We give details on these points and delineate further changes in our individual answers. For 3. and 5., we are currently running these experiments. If they are done before the end of the discussion period we will provide the additional results below, in the camera ready otherwise.

---

> ### Author Response · Authors · 2024-11-26
> **Additional experiments are running, partial results.**
>
> Our additional experiments are still running.
> - for 3. we performed clustering with GTE small. On both MMLU and ARC, we have training run that show results better than LSI, currently on par with sentence bert mini (LMs are mid training, 60B tokens). They might end up slightly better. We will report them in the paper on Figure 5 once they are done.
> - for 5., we tried cross-entropy difference for both MMLU and ARC. LMs are mid training (60B tokens) as well. The results on MMLU looks close to the classifier at the same point, The results on ARC are much worse than the classifier at this point. We still need to experiment with different entropy difference thresholds, an important hyperparameter for this method.

---

> > ### Author Response · Authors · 2024-12-02
> > **Most suggested experiments are done, results are provided below.**
> >
> > Most suggested experiments are done, results are provided below.
> > - For 5., *Cross-Entropy-Difference (CED)* has been evaluated as a baseline on Arc-E, Arc-C, MMLU.
> > Its results are better than base, and better than Doge on MMLU. CRISP is best, and Classifier is second best on the 3 datasets. CED is also the most expensive method, since scoring the whole redpj2 dataset with 2 LMs is more expensive than computing the redpj2 sentence BERT embeddings (main
> > source of selection cost for Classifier, CRISP).
> >
> > ```
> >            | Arc-E | Arc-C | MMLU  |
> > -----------|-------|-------|-------|
> > Base       |  58.4 |  27.4 |  30.0 |
> > CED        |  58.9 |  30.5 |  31.1 |
> > Doge       |  66.2 |  33.3 |  31.0 |
> > Classifier |  68.2 |  36.7 |  32.4 |
> > CRISP      |  71.2*|  38.6*|  33.4*|
> >
> > [This table extends Figure 4 and Table 10]
> > ```
> >
> >
> >
> > - For 3., GTE-small (Li et al, arXiv:2308.03281) has been evaluated as an *alternative representation* to sentence BERT all-MiniLM-L6-v2 (Reimers et al, arXiv:1908.10084) to perform the clustering of CRIPS. The results with the GTE-small representation are better than all-MiniLM-L6-v2 for 64,
> > 4096 clusters. The results over 262k are better or equal for SBERT all-MiniLM-L6-v2.
> > The results for 16m clusters are still running for the ARC datasets.
> > These additional experiments show that exploring further the (embedding x number of clusters) settings could further improve the results of CRISP.
> >
> > ```
> > Clusters | Emb.   | Arc-E | Arc-C | MMLU  |
> > ---------|--------|-------|-------|-------|
> > 64       | LSI    |  65.0 |  31.8 |  31.1 |
> >          | SBERT  |  65.2 |  33.0 |  31.2 |
> >          | GTE    |  67.8*|  33.8*|  31.5*|
> > ---------|--------|-------|-------|-------|
> > 4096     | LSI    |  67.6 |  35.4 |  31.6 |
> >          | SBERT  |  69.7 |  36.5 |  32.6 |
> >          | GTE    |  69.9*|  37.0*|  32.8*|
> > ---------|--------|-------|-------|-------|
> > 262k     | LSI    |  66.5 |  36.4 |  31.3 |
> >          | SBERT  |  71.2*|  38.6*|  33.4*|
> >          | GTE    |  69.3 |  37.6 |  33.4*|
> > ---------|--------|-------|-------|-------|
> > 16m      | LSI    |  53.8 |  29.6 |  29.0 |
> >          | SBERT  |  62.3 |  33.8 |  30.8 |
> >          | GTE    |Running|Running|  31.7*|
> >
> > [This table extends Figure 5]
> > ```
> >
> > We will finalize these results and add them to the paper.
> > They strengthen our empirical evaluation and we thank the reviewers for suggesting them.

---

### Meta-Review · Area_Chair_SuQb · 2024-12-21

**Metareview:**

This paper proposes a method called Clustered Importance Sampling (CRISP) for pre-training specialist LMs when domain-specific data is limited. CRISP adjusts the distribution of a generalist dataset to align with a limited specialist dataset by resampling the generalist data based on a cluster distribution. The authors demonstrate that CRISP improves performance on language modeling and multiple-choice question accuracy tasks compared to other adaptation techniques.

The authors show that the method achieves competitive performance compared to other pre-training methods on various tasks and domains. The paper provides a comprehensive analysis of the method, exploring the effects of various hyperparameters and training configurations. The method is shown to be scalable to different model sizes.

The reviewers argue that the paper does not provide clear guidance on determining the optimal number of clusters, which is crucial for CRISP's performance. CRISP offers limited additional benefits for multiple-choice question tasks after fine-tuning compared to task-specific pre-training. When the specialist dataset is very small, CRISP appears to be less effective in enhancing end-task accuracy.

Overall, this paper presents a promising method for pre-training specialist LMs with limited data. Despite some limitations which the authors have agreed to acknowledge, I think this is a good paper and I recommend acceptance.

**Additional Comments On Reviewer Discussion:**

Reviewer RQp5 and Z9XR raised concerns about the novelty of the method and suggested that the authors should more explicitly highlight their contributions compared to previous work. Reviewer oNof suggested that the authors provide more details on the computational cost of CRISP. Reviewer TBtV proposed comparing against alternative baseline methods

Additionally, the reviewers suggested several potential improvements to the paper:

* Providing more details on how to determine the optimal number of clusters for CRISP.
* Try alternative embedding methods for clustering
* Include other baselines like Cross-Entropy difference.
* Discussing the limitations of CRISP, such as its potential for diminishing returns with very small specialist datasets.

The authors have either provided these results in the rebuttal (and the reviewers have raised scores) or indicated that they are willing to address these suggestions in a revised version of the paper.

---

### Decision · Program_Chairs · 2025-01-22

Accept (Poster)